# Cancer stage at presentation for incarcerated patients at a single urban tertiary care center

**Kathryn I. Sunthankar**[1☯], **Kevin N. Griffith**[2☯], **Stephanie D. Talutis**[3], **Amy K. Rosen**[4], **David B. McAneny**[3], **Matthew H. Kulke**[3], **Jennifer F. Tseng**[3], **Teviah E. Sachs**[3]*

**1** Department of Medicine, Vanderbilt University Medical Center, Nashville, TN, United States of America, **2** Boston University School of Public Health, Boston, MA, United States of America, **3** Boston University School of Medicine, Boston, MA, United States of America, **4** Center for Healthcare Organization and Implementation Research, VA Boston Healthcare System, Boston, MA, United States of America

☯ These authors contributed equally to this work.
* teviah.sachs@bmc.org

**Data Availability Statement:** We understand the importance of making the data available, however, as incarcerated patients are a protected group, we have consulted with our IRB and determined that

## Abstract

Patients who are incarcerated are a vulnerable patient population and may suffer from less access to routine cancer screenings compared to their non-incarcerated counterparts. Therefore, a thorough evaluation of potential differences in cancer diagnosis staging is needed. We sought to examine whether there are differences in cancer stage at initial diagnosis between non-incarcerated and incarcerated patients by pursuing a retrospective chart review from 2010–2017 for all patients who were newly diagnosed with cancer at an urban safety net hospital. Incarceration status was determined by insurance status. Our primary outcome was incarceration status at time of initial cancer diagnosis. Overall, patients who were incarcerated presented at a later cancer stage for all cancer types compared to the non-incarcerated (+.14 T stage, p = .033; +.23 N stage, p < .001). Incarcerated patients were diagnosed at later stages for colorectal (+0.93 T stage, p < .001; +.48 N stage, p < .001), oropharyngeal (+0.37 N stage, p = .003), lung (+0.60 N stage, p = .018), skin (+0.59 N stage, p = 0.014), and screenable cancers (colorectal, prostate, lung) as a whole (+0.23 T stage, p = 0.002; +0.17 N stage, p = 0.008). Incarcerated patients may benefit from more structured screening protocols in order to improve the stage at presentation for certain malignancies.

## Introduction

Persons who are incarcerated represent a unique and vulnerable, patient population. During the time they are incarcerated, these patients receive variable access to health care that is dependent upon the state in which they are incarcerated, the length of their incarceration, and the arrangements for health care providers made by their institution [1]. Incarcerated patients (IP) face both systemic barriers to care as well as vulnerability due to impediments to self-advocating due to their status as an inmate [2]. Both these issues may delay or limit their receipt of treatments that–outside the prison system–are more often given, such as direct-acting anti-viral agents for hepatitis C infection and opioid maintenance therapy for substance

the data may, in combination, become identifying (IRB #H-35743). Therefore, we believe it is more appropriate to make the data available upon request, rather than fully available to the public. Data are available from the Boston Medical Center Institutional Data Access for researchers who meet the criteria for access to confidential data. They can be reached by phone (617-358-7383) or email (ohra@bu.edu).

**Funding:** The author(s) received no specific funding for this work.

**Competing interests:** The authors have declared that no competing interests exist.

use disorders [3, 4]. Additionally, in the United States, IP are less likely to receive screenings for certain cancers in a timely fashion [5]. Overall mortality in persons with a history of incarceration is higher compared to people who have never been incarcerated [6, 7]. Additionally, IP carry a higher burden of chronic and acute health conditions, particularly infections including HIV, hepatitis C virus and other sexually transmitted infections [8–10]. Together, the barriers in access to care, in combination with increased chronic medical conditions and decreased screening tests, are likely associated with and may directly contribute to the increased overall mortality in people who are currently incarcerated or have a history of incarceration [11, 12].

Previous research has demonstrated that cancer is a leading cause of death for those who are currently or formerly incarcerated. A retrospective cohort study of 4,026 Texas state prisoners showed cancer and cardiovascular disease were the leading causes of death in prisoners age 55–84 [13]. Another similar study of 797 North Carolina prisoners showed cancer as the second leading cause of death. This study revealed that IP had higher liver cancer-related mortality compared to non-incarcerated population [14]. This difference does not necessarily end upon release from prison, as previously incarcerated patients have an excess mortality due to cancer compared to never-incarcerated patients [15]. These studies underscore the importance of recognizing cancer in IP, not just for their health while incarcerated, but also after release from prison.

The etiology of increased mortality from cancer in IP has not been subjected to rigorous study and is poorly understood. HCV and tobacco use, known cancer risk factors, are more prevalent amount IP populations. Taking these into account, it is not surprising that the incarcerated population has an increased incidence of both lung cancer and hepatocellular cancer compared to patients who are not incarcerated [16–18]. Current screening recommendations from the United States Preventive Services Task Force (USPSTF) for lung cancer include low dose Computerized Tomography (CT) scan for patients over 55 with more than 15 pack year smoking history [19]. Screening for hepatocellular carcinoma is indicated in patients with cirrhosis due to hepatitis C or alcohol and involves frequent abdominal imaging (either ultrasound or CT) according to society guidelines [20]. USPSTF recommends screening for colorectal cancer in all people age 50 and older [21]. Despite increased incidence of smoking, hepatitis C and alcoholism, appropriate screenings for hepatocellular carcinoma, lung cancer or colorectal cancer are not always performed according to national and international guidelines in the IP population [5]. Multiple reasons for decreased screenings among patients who are currently or previously incarcerated have been suggested, including poorer connection with primary care resources, lack of availability while incarcerated and following incarceration due to systemic barriers patients who were previously incarcerated also face [22, 23]. Regarding patients who are currently incarcerated, a recent study of the incarcerated population in Ontario, Canada showed these patients are more likely to be overdue for breast and colorectal cancer screening [11]. While not previously studied or documented, decreased availability of screening in prison is likely to also contribute to the lack of screening in this population.

Together these studies have highlighted that IP are less likely to receive the appropriate screenings compared to the general population. Therefore, this already vulnerable population is often not aware of that for which they must self-advocate nor are they in a position to do so. These situations, compounded, further decrease the likelihood of this at-risk population having their cancers detected at an early stage. In our study we assessed the association between incarceration and cancer stage upon initial diagnosis in all cancers as well as specific cancer sub-types, including those with robust screening guidelines.

## Materials and methods

### Data and population

This retrospective, observational study was carried out with a patient cohort at a large, urban, tertiary care safety-net hospital in New England from January 1, 2010 until December 31, 2017. Patients were identified through the hospital's cancer registry database, with incarceration status determined through administrative billing data. After excluding non-malignant lesions, a total of 116 incarcerated and 2,860 non-incarcerated patients were included for analysis. After review of each case, additional cases were excluded due to either cancers not staged by American Joint Committee on Cancer (AJCC) 7th edition staging system or if the total number of cancer subtypes was too small for robust statistical analysis. A total of 74 incarcerated and 1,408 non-incarcerated patients were included in the final analysis (S1 Fig). Of these, 100% IP cases and 160 of non-incarcerated patients (NIP) cases (11.3% of total) were reviewed to ensure internal consistency of the cancer registry database. Patients were excluded if they carried a prior diagnosis of the primary cancer (i.e. previously treated and seeking second opinion) or had a recurrence of a prior cancer. Patients were marked as having a previous diagnosis of cancer if they were either diagnosed prior to January 1, 2010 and still receiving treatment throughout the study period or if they were diagnosed at an outside hospital and received treatment (surgery, radiation or chemotherapy) before transferring their care to our institution. Patients were excluded upon the basis of prior cancer recurrence if they had a relapse within 5 years of their last curative therapy. If there was a recurrence that occurred outside of the 5 years, the patient was included in the analysis as having a new, de novo cancer. These exclusion criteria allowed us to evaluate the staging of cancer upon initial diagnosis for non-incarcerated and incarcerated patients seen at our institution. This study was approved and monitored by the hospital's Institutional Review Board. Given the study was retrospective, informed consent was waived by the IRB.

### Study variables

Risk factors such as patients' age at diagnosis, race and gender were extracted from the cancer registry database. Additional risk factor data for patients with lung or hepatocellular carcinoma were extracted manually via chart review. For lung cancer, extracted data included occupational exposures (radon, asbestos), TB status, smoking history and family history. The initial history and physical note at diagnosis of cancer was reviewed for each of these risk factors. Patient's smoking history was documented in pack years. For patients who had quit, their year of quitting was also noted. TB status was noted as positive or negative based on either PPD, Quant-GOLD assay or three AFB negative sputum.

For hepatocellular carcinoma, history of hepatitis B and C viruses, cirrhosis, alcohol abuse and smoking history were extracted as important risk factors. Patients were noted to have a history of HCV if they had a recent serum antibody test positive for anti-HCV. HBV status was noted based upon routine hepatitis panel as follows: current infection with +HBsAg, resolved infection with -HBsAg, +anti-HBc, +anti-HBs and never infected with -HBsAg and -anti-HBc. Patients with cirrhosis or alcohol abuse were evaluated based upon ICD-9 or ICD-10 codes for each of the conditions.

The AJCC 7th edition staging system was used to stage each patient's cancer at time of diagnosis [24]. We used the clinical tumor (T), nodal (N), metastatic (M) and full clinical AJCC stage as outcome variables indicative of prognosis [cite]. Additionally, we used binary indicators of tumor staging by grouping T1 and T2 as early stage and T3 and T4 as later stage. Similarly, for nodal stage N0 was grouped as early while N1, N2 and N3 were late. For clinical

AJCC, stages I and II were marked as early stage and stages III and IV were marked as later stage.

## Analytic approach

Our analysis proceeded in three steps. We first characterized the differences between the groups using two-tailed t-tests or chi-squared tests as appropriate. We then calculated mean cancer outcomes for IP and non-IP and compared them using two-sided t-tests, showing potential differences in cancer staging before adjustment for observed risk factors. Next, we employed inverse probability of treatment weighting (IPTW) to identify the effect of incarceration on initial cancer staging. This method is increasingly used in cancer studies, allowing investigators to reduce bias when assessing the effects of an intervention when treatment and control groups are nonequivalent [25].

For the IPTW, propensity scores were obtained using logistic regression with incarceration as the outcome and age, race, and gender as predictors. Each subject's weight was defined as the inverse of the probability of receiving the treatment (or non-treatment) that the subject received; incarceration vs. non-incarceration.

$$w_i = \frac{T_i}{\widehat{P}_i} + \frac{1 - T_i}{1 - \widehat{P}_i} \qquad \text{If treated, } w_i = \frac{1}{\widehat{P}_i} \quad \text{If control, } w_i = \frac{1}{1 - \widehat{P}_i}$$

Weighted linear regression models were then used to identify the effect of incarceration on the cancer outcomes, after controlling for other risk factors. The results may be interpreted as average changes in cancer staging, or changes in predicted probability of late treatment for dichotomized outcomes. When the propensity score model is correctly specified, this technique consistently estimates the true treatment effect [26].

## Results

### Patient characteristics and cancer incidence

We identified a total of 74 patients who were diagnosed with cancer at our institution while incarcerated (IP) and 1408 patients who were not incarcerated (NIP) during the study period. The characteristics of our study sample are listed in Table 1. Demographic differences between the two groups were assessed using two-tailed t-tests or chi-squared tests as appropriate. Compared to NIP, IP were more likely to be male (94.6% vs. 68.8%, p < 0.001) and an older median age (62.4 vs. 57.2 years, p<0.001). In both IP and NIP, Caucasian (67.6% vs. 67.0%) was the most common race followed by African American (27.0% vs. 28.0%) and Hispanic (5.4% vs. 5.0%), revealing similar racial balance between groups (p = 0.978). The prevalence of specific cancers, however, varied between groups (p < .001), with hepatobiliary (31.1%), bronchopulmonary (20.3%) and oropharyngeal (14.9%) being most common in IP, while oropharyngeal (24.9%), bronchopulmonary (22.3%) and prostate (21.1%) were most common in NIP (Fig 1, S1 Table).

### Unadjusted disparities in cancer staging

Evaluating AJCC staging for all cancers combined, IP were diagnosed at slightly later T (2.37 vs. 2.31, p = 0.618) and N stages (0.72 vs. 0.67, p = 0.670), however these differences did not reach significance (Table 2, S2 Table). IP and NIP were diagnosed at approximately the same M and overall AJCC stages.

As the cancer types differed significantly between the IP and NIP groups, we next sought to evaluate the staging within screening cancers and individually for each cancer type. While

**Table 1. Summary demographics for patients included in analysis.**

| Variable | | Incarceration Status | | P-value |
|---|---|---|---|---|
| | | **No** | **Yes** | |
| N | | 1,408 | 74 | – |
| Age in years (Mean (SD)) | | 57.2 (10.3) | 62.4 (27.8) | < .001 |
| Sex (N (%)) | | | | < .001 |
| | Male | 968 (68.9) | 70 (94.6) | |
| Race (N (%)) | | | | 0.895 |
| | African American | 394 (28.0) | 20 (27.0) | |
| | Caucasian | 943 (67.0) | 50 (67.6) | |
| | Hispanic | 71 (5.0) | 4 (5.4) | |
| Tumor stage (%) | | | | 0.287 |
| | 0 | 14 (1.0) | 1 (1.4) | |
| | 1 | 417 (30.5) | 16 (21.9) | |
| | 2 | 391 (28.6) | 25 (34.2) | |
| | 3 | 228 (16.7) | 17 (23.3) | |
| | 4 | 317 (23.2) | 14 (19.2) | |
| Nodal stage (N (%)) | | | | 0.254 |
| | 0 | 853 (62.1) | 45 (62.5) | |
| | 1 | 195 (14.2) | 8 (11.1) | |
| | 2 | 254 (18.5) | 13 (18.1) | |
| | 3 | 72 (5.2) | 6 (8.3) | |
| Metastatic stage (N (%)) | | | | 0.545 |
| | 0 | 1099 (79.5) | 59 (80.8) | |
| | 1 | 283 (20.5) | 14 (19.2) | |
| AJCC stage (N (%)) | | | | 0.507 |
| | 0 | 6 (0.4) | 1 (1.4) | |
| | 1 | 300 (21.7) | 12 (16.2) | |
| | 2 | 369 (26.7) | 25 (33.8) | |
| | 3 | 217 (15.7) | 11 (14.9) | |
| | 4 | 491 (35.5) | 25 (33.8) | |

Differences were assessed using either two-tailed t-tests (for continuous variables) or chi-squared tests (for categorical variables). American Joint Committee on Cancer (AJCC).

many individual cancer types did not show differences in overall staging or sub-staging, the IP group was diagnosed at a T stage that was 0.92 higher than the NIP group for colorectal cancers (95% CI, 0.22 to 1.62). Additionally, there were slight differences in M staging for oropharyngeal and skin cancers that were statistically significant but due to small numbers, were not considered clinically significant.

To further evaluate our patient population, we grouped each patient into early and late stage for tumor (T1, T2 vs. T3, T4), nodal (N0 vs. N1, N2, N3) and AJCC clinical stage (I, II vs. III, IV) groups to form binary indicators of early and late stage. There were no significant differences in early and late stage tumor, nodal or AJCC clinical staging between the IP and NIP groups, both overall or for specific cancers (S3 Table). IP were more likely to be diagnosed at a late stage for colorectal cancer (86% vs. 52%, p = 0.055), but this result was marginally significant.

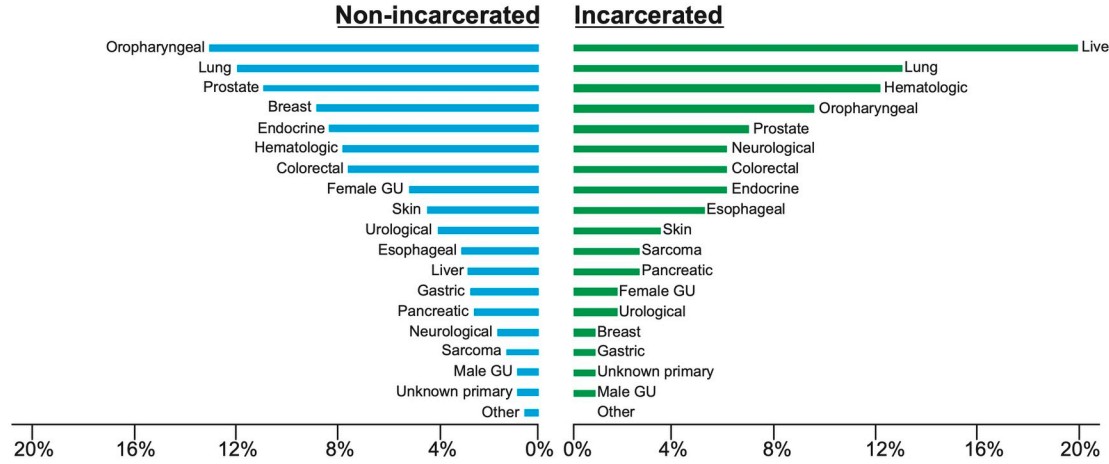

**Fig 1. Incidence of different cancer subtypes in incarcerated patients and non-incarcerated patients.** Bars represent the percentage of each cancer subtype in IP and NIP cases. Hepatobiliary (31.1%), Bronchopulmonary (20.3%) and oropharyngeal (14.9%) were most common in IP, while oropharyngeal (24.9%), bronchopulmonary (22.3%) and prostate (21.0%) were most common in NPI.

## Inverse probability of treatment weighting

The IPTW results for cancer staging are contained in Table 3. Overall, IP received cancer diagnoses that were at late T stages (+0.14, 95% CI 0.01 to 0.26) and N stages (+0.23, 95% CI 0.13 to 0.34) when compared to those who were not incarcerated (S4 Table). Similar results were found when focusing on screenable cancers only, where IP were diagnosed at later T stages (+0.23, 95% CI 0.09 to 0.38) and N stages (+0.17, 95% CI 0.04 to 0.29). For individual cancer types, the largest disparity for IP vs NIP occurred with T staging for colorectal cancer (+0.93, 95% CI 0.59 to 1.28). IP also received higher initial N staging for oropharyngeal (+0.37, 95% CI 0.12 to 0.61), lung (+0.60, 95% CI 0.11 to 1.10), colorectal (+0.48, 95% CI 0.22 to 0.73) and skin cancers (+0.59, 95% CI 0.12 to 1.05). On average, IP also received higher overall staging compared to NIP but these differences were only significant for oropharyngeal cancers (+0.36, 95% CI 0.05 to 0.67).

Fewer differences were observed in the IPTW analyses comparing absolute rates of early vs. late diagnoses (S5 Table). Compared to NIP, IP were more likely to receive a late T stage diagnosis for screenable cancers (+0.07, 95% CI 0.03 to 0.14) and colorectal cancer (+0.34, 95% CI 0.19 to 0.49). For N staging, IP were more likely to be diagnosed late for oropharyngeal (+0.22, 95% CI 0.10 to 0.34) and colorectal cancers (+0.16, 95% CI 0.01 to 0.31).

Since patients who are incarcerated are more likely to harbor risk factors for certain cancers (S6 and S7 Tables), we performed IPTW analyses which accounted for these risk factors for

**Table 2. Descriptive statistics of cancer staging by incarceration status.**

| Cancer Type | Clinical Stage | | | | | | | | | | | |
| --- | --- | --- | --- | --- | --- | --- | --- | --- | --- | --- | --- | --- |
| | T | | Diff | N | | Diff | M | | Diff | AJCC | | Diff |
| | Incarcerated | | | Incarcerated | | | Incarcerated | | | Incarcerated | | |
| | Yes | No | | Yes | No | | Yes | No | | Yes | No | |
| Oropharyngeal | 2.73 | 2.56 | 0.17 | 1.36 | 0.92 | 0.45 | 0.00 | 0.02 | -0.02* | 3.45 | 3.00 | 0.46 |
| Lung | 2.67 | 2.52 | 0.15 | 1.13 | 1.22 | -0.08 | 0.47 | 0.46 | 0.00 | 2.87 | 2.96 | -0.09 |
| Liver | 2.17 | 1.97 | 0.20 | 0.14 | 0.18 | -0.05 | 0.09 | 0.19 | -0.11 | 2.30 | 2.21 | 0.10 |
| Esophageal | 1.67 | 2.26 | -0.60 | 0.83 | 0.72 | 0.11 | 0.33 | 0.31 | 0.02 | 2.50 | 2.73 | -0.23 |
| Colorectal | 3.29 | 2.37 | 0.92* | 1.14 | 0.57 | 0.57 | 0.29 | 0.31 | -0.03 | 2.86 | 2.58 | 0.27 |
| Prostate | 1.75 | 1.94 | -0.19 | 0.12 | 0.13 | 0.00 | 0.12 | 0.11 | 0.01 | 2.38 | 2.39 | -0.01 |
| Skin | 2.00 | 2.02 | -0.02 | 1.00 | 0.25 | 0.75 | 0.00 | 0.04 | -0.04* | 1.75 | 1.60 | 0.15 |
| Screenable | 2.46 | 2.30 | 0.16 | 0.69 | 0.67 | 0.02 | 0.19 | 0.21 | -0.02 | 2.78 | 2.63 | 0.15 |
| Overall | 2.37 | 2.31 | 0.06 | 0.72 | 0.67 | 0.05 | 0.19 | 0.20 | -0.01 | 2.64 | 2.64 | -0.01 |

The table displays unadjusted averages and differences in tumor staging between prisoners and non-prisoners. Screenable cancers include liver, lung, colorectal, and prostate. Differences were assessed using two-sided t-tests.

*$p < 0.05$

**$p < 0.01$

***$p < 0.001$.

two cancer types where such data were available: hepatocellular carcinoma (HCC) and lung (S4 and S5 Tables). Incorporating the risk factors for HCC (viral status, smoking history, history of cirrhosis, history of alcoholism) had no impact on the results for stage of diagnosis. However, incorporating the risk factors for lung cancer (smoking history, asbestos exposure) showed a significant increase in nodal stage (+0.60, $p = 0.05$) in IP as compared to NIP (S4 Table).

## Discussion

Patients who are incarcerated experience an increased mortality due to cancer, however the mechanism behind this has remained unclear. Our findings comport with previous research,

**Table 3. Adjusted regression results for the effects of incarceration status on initial cancer staging, by cancer subtype.**

| Cancer Type | Incarcerated (#) | | Clinical Stage | | | |
| --- | --- | --- | --- | --- | --- | --- |
| | No | Yes | T | N | M | AJCC |
| Oropharyngeal | 351 | 11 | 0.16 | 0.37** | -0.01 | 0.36* |
| Lung | 314 | 15 | 0.19 | 0.03 | 0.06 | 0.03 |
| Liver | 67 | 23 | 0.21 | -0.03 | -0.07 | 0.11 |
| Esophageal | 70 | 6 | -0.39 | 0.25 | 0.05 | 0.07 |
| Colorectal | 198 | 7 | 0.93*** | 0.48*** | -0.06 | 0.14 |
| Prostate | 296 | 8 | -0.12 | 0.02 | 0.04 | 0.04 |
| Skin | 112 | 4 | 0.22 | 0.59* | -0.06 | 0.06 |
| Screenable | 875 | 53 | 0.23** | 0.17** | -0.04 | 0.04 |
| Overall | 1408 | 74 | 0.14* | 0.23*** | -0.04 | 0.09 |

The table displays average differences in tumor staging between prisoners and non-prisoners after inverse probability of treatment weighting. Screenable cancers include liver, lung, colorectal, and prostate. $p < 0.05$

**$p < 0.01$

***$p < 0.001$.

which has suggested differential health literacy and screening rates by incarceration status as etiologies of increased mortality from cancer. We characterized differences in cancer stage at diagnosis between patients who are incarcerated and those who are not incarcerated. Our institutional setting is particularly suited for this analysis due to its treatment of a large portion of the state's incarcerated patients, and its status as an urban safety net hospital with NIP demographics similar to the IP group.

Our initial analysis examined unadjusted cancer staging between the IP and NIP groups. While we found that colorectal cancers were diagnosed at a later T stage, the populations were generally similar with respect to cancer staging. However, the IP group was younger and more likely to be male compared to the NIP group. Therefore, we accounted for the demographic factors using IPTW models.

After adjusting for demographics, we found overarching disparities in T, N, and AJCC clinical stage between the IP and NIP groups. IP were diagnosed at later stages in our overall analysis as well as for several specific cancer types including colorectal, oropharyngeal, lung, skin. Screenable cancers (liver, lung, colorectal, prostate) showed a statistically significant increase in T and N staging for IP, suggesting these disparities in staging could be driven by lower screening rates and thus less early detection.

Our study suggests opportunities to improve screening and care processes for certain incarcerated patients. At a population level, the observed disparities in cancer staging between IP and NIP represent a potential human cost in terms of both quality of life and expected survival post-diagnosis. Later diagnoses may also limit the ability of IP to gain employment or assimilate back into their communities following release. The most striking difference is with the patients with colorectal cancer as our data indicates they are diagnosed at a significantly later stage. Guidelines and recommendations for colorectal cancer screening with colonoscopy have been established for decades and screening has an impressive effect on early detection of cancer as well as prevention with polypectomy. Understanding that IP are diagnosed at later stages should inform health care practices within the prison system and direct routine screening of all eligible patients. We recognize that cost is likely to represent a significant barrier and while colonoscopy is often preferred, additional screening through fecal occult blood tests are also viable options for this population and could aid in earlier detection. Current practices for screening in our state's prison system are unknown, however it is notable that none of the patients diagnosed with colorectal cancer or lung cancer had documented appropriate screenings while incarcerated or were referred to our hospital following a positive result from screening.

While we detected differences in stage at diagnosis for multiple screenable cancers, it must also be recognized that many cancers do not have established screening tests, including oropharyngeal and skin cancers. The differences in stage at diagnosis in these cancers could be rooted in at least two possibilities: 1) decreased access to care overall while incarcerated or 2) life-long systemic injustice of patients who are of low socioeconomic status and make up the majority of the incarcerated persons in the United States. It must also be recognized that poor access health care is not only found in our prison system. It is perpetuated throughout each stage of life and leads to unstable households, adverse childhood events, unemployment, mental illness, substance use and acquisition of habits that increase risk for cancer (smoking, alcohol use, sexually transmitted infections, injection drug use and its associated infections). Therefore, we must promote screening of cancers in our incarcerated population but also recognize that prevention of patients becoming incarcerated starts with adequate access to health care early in life as well.

Our study has important strengths. Our institutional setting was particularly suited for this analysis due to its treatment of a large portion of the state's incarcerated patients, and its status as an urban safety net hospital with NIP demographics similar to the IP group. However, our

study also has several limitations. Due to the observational design, our findings should be interpreted as associations and we cannot make causal inferences between incarceration and later stage of cancer diagnosis. Certain demographic information was not available, including zip codes and income for the non-incarcerated group, which affected our ability to adjust for socioeconomic status. Our sample size limited our ability to conduct subgroup analyses for certain cancer types. Lastly, the continuity of care in the correctional system is fragmented and once patients are released they are often lost to follow up. Additionally, the length of incarceration for each patient is different. Taken both of these together we were thus unable to analyze the effect of incarceration (occurrence or length) on cancer survival and mortality. Further studies are needed to evaluate the impact of delayed detection in IP on future health outcomes, especially for colorectal cancers.

## Conclusions

Compared to those who are not incarcerated, incarcerated persons may present at a later stage for several cancer types. This disparity is especially evident in screenable cancers such as colorectal cancer. Our results suggest a potential for improvement in the screening protocols for those who are incarcerated, particularly for colorectal cancers. Given the many barriers associated with routine screening for all who are incarcerated, initial reforms might be best focused on high-risk individuals.

## Supporting information

**S1 Fig. Construction of the study sample.**
(DOCX)

**S1 Table. Frequency of cancer subtypes in incarcerated and non-incarcerated population.**
(DOCX)

**S2 Table. Descriptive statistics of cancer staging by incarceration status with confidence intervals.**
(DOCX)

**S3 Table. Descriptive statistics for early vs. late cancer staging, by cancer subtype.**
(DOCX)

**S4 Table. Adjusted regression results for the effects of incarceration status on initial cancer staging, by cancer subtype with confidence intervals.**
(DOCX)

**S5 Table. Adjusted regression results for the effects of incarceration status on early vs. late cancer staging, by cancer subtype.**
(DOCX)

**S6 Table. Lung cancer risk factors in IP and NIP groups.**
(DOCX)

**S7 Table. Hepatocellular carcinoma risk factors in IP and NIP groups.**
(DOCX)

## Acknowledgments

We would like to thank our colleagues from the Boston School of Public Health who provided insight and expertise that greatly assisted our research.

## Author Contributions

**Conceptualization:** Kathryn I. Sunthankar, Amy K. Rosen, Teviah E. Sachs.

**Data curation:** Kathryn I. Sunthankar.

**Formal analysis:** Kathryn I. Sunthankar, Kevin N. Griffith.

**Investigation:** Kathryn I. Sunthankar, Teviah E. Sachs.

**Methodology:** Kevin N. Griffith, Teviah E. Sachs.

**Project administration:** Teviah E. Sachs.

**Resources:** Teviah E. Sachs.

**Supervision:** Teviah E. Sachs.

**Writing – original draft:** Kathryn I. Sunthankar, Kevin N. Griffith.

**Writing – review & editing:** Kathryn I. Sunthankar, Kevin N. Griffith, Stephanie D. Talutis, Amy K. Rosen, David B. McAneny, Matthew H. Kulke, Jennifer F. Tseng, Teviah E. Sachs.

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
