## [Decision Letter · Decision Letter 0]

11 Jun 2020

PONE-D-19-34040

Stage at presentation for incarcerated patients at a single urban tertiary care center

PLOS ONE

Dear Dr. Sunthankar,

Thank you for submitting your manuscript to PLOS ONE. After careful consideration, we feel that it has merit but does not fully meet PLOS ONE’s publication criteria as it currently stands. Therefore, we invite you to submit a revised version of the manuscript that addresses the points raised during the review process.

We look forward to receiving your revised manuscript.

Kind regards,

Sungwoo Lim, DrPH

Academic Editor

PLOS ONE

Journal Requirements:

2. In ethics statement in the manuscript and in the online submission form, please provide additional information about the patient records used in your retrospective study. Specifically, please ensure that you have discussed whether all data were fully anonymized before you accessed them and/or whether the IRB or ethics committee waived the requirement for informed consent. If patients provided informed written consent to have data from their medical records used in research, please include this information.

Reviewers' comments:

Reviewer's Responses to Questions

**Comments to the Author**

1. Is the manuscript technically sound, and do the data support the conclusions?

Reviewer #1: No

Reviewer #2: Yes

2. Has the statistical analysis been performed appropriately and rigorously? 

Reviewer #1: Yes

Reviewer #2: I Don't Know

3. Have the authors made all data underlying the findings in their manuscript fully available?

Reviewer #1: Yes

Reviewer #2: Yes

4. Is the manuscript presented in an intelligible fashion and written in standard English?

Reviewer #1: Yes

Reviewer #2: Yes

5. Review Comments to the Author

Reviewer #1: This analysis does not really have enough patients to make any reliable conclusions about this population. Also not having the insurance or income status is very problematic since this population of is often in the lower socioeconomic stratus which results in patients not doing screening and having later stage cancers.

Reviewer #2: This is an extremely welcome addition to the literature. I have several comments which, I believe, can make it a bit more accessible and therefore, important as the basis of advocacy for this population.

Page 3, line 71—“in some cases”? Please, this is as vulnerable population. Why are you mincing words on this?

Page 3, line 74—Please add something to the effect of the “health care provider arrangements with which the facility contracts”. The type of provider can make a huge difference, especially when contracts are with for-profit companies with incentives for minimal care.

Page 2, line 104—these recommendations need to be summarized—some are more and some much less available, or even recommended.

Page 4, line 108—why only blame the patient here? At least mention that one possible reasons for decreased screening is the lack of availability by the prison health facility!

Page 4, line 114—awkward sentence

Page 5, line 128-9—do you mean the number of specific type of cancer? Up to now, you have not mentioned that you are going to analyze types of cancer so this was confusing

Page 9, Table 1—spell out Tumor, Nodal, Metastatic; give a footnote for complete name of AJCC.

Page 14, line 302—this point needs much more emphasis

Overall Discussion—There needs to be more discussion about cancers that have and do not have reliable screening tests available, including tests that move beyond general screening (colorectal cancer) and look at people with, say HepC—how should their increased risk be addressed. I appreciate that this is not a medical paper but that does not mean that oropharyngeal cancer (for which there is no screening test as far as I am aware) should be equated with colorectal (which has a cheap, simple fecal-occult blood test).

Given the above, I don’t see how you can possible avoid at least some discussion about the health care available to this population. Is screening even available? Is any education provided? Don’t blame this on patient literacy—providers of this population (who have NO choices) have some responsibility for health education. How much does screening cost?

A follow-up study might be to determine if the prisons in the state have any cancer screening guidelines and how these are accessed.

6. PLOS authors have the option to publish the peer review history of their article (what does this mean?). If published, this will include your full peer review and any attached files.

Reviewer #1: No

Reviewer #2: Yes: Patricia J Kelly

---

## [Author Response · Author response to Decision Letter 0]

29 Jun 2020

Reviewer #1: This analysis does not really have enough patients to make any reliable conclusions about this population. Also not having the insurance or income status is very problematic since this population of is often in the lower socioeconomic stratus which results in patients not doing screening and having later stage cancers.

We thank Reviewer #1 for their input. Our data compare patients with a new diagnosis of malignancy who are incarcerated to those who are non-incarcerated at our institution, which is an urban, high volume, safety net hospital. The majority of our patient population is minority underserved and represents a similar demographic to those who are incarcerated. As the only major safety-net hospital in the Boston metropolitan area, all patients are able to receive care through the MassHealth program, which is able to provide insurance to 95% or more of the otherwise uninsured persons. Therefore, we believe that the absence of insurance and income status would not play a significant role in our analysis. Prior published analyses of our patient population have shown no difference in oncologic outcomes between patients based on insurance status, race or ethnicity or income level at our institution. (see: Morgan et al, JOGS; PMID# 30097966 and Sridhar et al, JACS; PMID# 31212101)

Reviewer #2: This is an extremely welcome addition to the literature. I have several comments which, I believe, can make it a bit more accessible and therefore, important as the basis of advocacy for this population.

Page 3, line 71 (42) — “in some cases”? Please, this is as vulnerable population. Why are you mincing words on this?

We thank Reviewer #2 for their input and we agree. This has been removed. 

Page 3, line 74 (45) — Please add something to the effect of the “health care provider arrangements with which the facility contracts”. The type of provider can make a huge difference, especially when contracts are with for-profit companies with incentives for minimal care.

We thank Reviewer #2 for their input and we have revised the manuscript to state: During the time they are incarcerated, these patients receive variable access to health care that is dependent upon the state in which they are incarcerated, the length of their incarceration, and the arrangements for health care providers made by their institution.

Page 2, line 104 (74) —these recommendations need to be summarized—some are more and some much less available, or even recommended.

We thank Reviewer #2 for their input and we have revised the manuscript to state: Current screening recommendations from the United States Preventive Services Task Force (USPSTF) for lung cancer include low dose Computerized Tomography (CT) scan for patients over 55 with more than 15 pack year smoking history. Screening for hepatocellular carcinoma is indicated in patients with cirrhosis due to hepatitis C or alcohol and involves frequent abdominal imaging (either ultrasound or CT) according to society guidelines. USPSTF recommends screening for colorectal cancer in all people age 50 and older. Despite increased incidence of smoking, hepatitis C and alcoholism, appropriate screenings for hepatocellular carcinoma, lung cancer or colorectal cancer are not always performed according to national and international guidelines in the IP population [4].

Page 4, line 108 (84) — why only blame the patient here? At least mention that one possible reasons for decreased screening is the lack of availability by the prison health facility!

We thank Reviewer #2 for their input and we have revised the manuscript to state: “While not previously studied or documented, decreased availability of screening in prison is likely to also contribute to the lack of screening in this population.” 

Page 4, line 114 (91) —awkward sentence

We thank Reviewer #2 for their input and we have revised the manuscript to state: “Together these studies have highlighted that IP have lower health literacy regarding screenings and they are unlikely to receive the appropriate screenings while incarcerated. Therefore, this already vulnerable population is often not aware of that for which they must self-advocate nor are they in a position to do so.

Page 5, line 128-9 (108) —do you mean the number of specific type of cancer? Up to now, you have not mentioned that you are going to analyze types of cancer so this was confusing

We thank Reviewer #2 for their input and we have revised the manuscript to state: “cancer subtypes”. We have also adjusted the last sentence of the Introduction (line 98) to state: “In our study we assessed the association between incarceration and cancer stage upon initial diagnosis in all cancers as well as specific cancer sub-types, including those with robust screening guidelines.”

Page 9, Table 1—spell out Tumor, Nodal, Metastatic; give a footnote for complete name of AJCC.

We thank Reviewer #2 for their input and we have revised the manuscript as recommended.

Page 14, line 302 (284) —this point needs much more emphasis. 

We thank Reviewer #2 for their input and we have revised the manuscript to state: “The most striking difference is with the patients diagnosed with colorectal cancer as our data indicates they are diagnosed at a significantly later stage. Guidelines and recommendations for colorectal cancer screening with colonoscopy have been established for decades and screening has an impressive effect on early detection of cancer as well as prevention with polypectomy. Understanding that IP are diagnosed at later stages should inform health care practices within the prison system and encourage routine screening of all eligible patients. We recognize that cost is likely to represent a significant barrier and while colonoscopy is often preferred, additional screening through fecal occult blood tests are also viable options for this population and could aid in earlier detection.”

Overall Discussion—There needs to be more discussion about cancers that have and do not have reliable screening tests available, including tests that move beyond general screening (colorectal cancer) and look at people with, say HepC—how should their increased risk be addressed. I appreciate that this is not a medical paper but that does not mean that oropharyngeal cancer (for which there is no screening test as far as I am aware) should be equated with colorectal (which has a cheap, simple fecal-occult blood test).

We thank Reviewer #2 for their input and we have revised the manuscript to state: “While we detected differences in stage at diagnosis for multiple screenable cancers, it must also be recognized that many cancers do not have established screening tests, including oropharyngeal and skin cancers. The differences in stage at diagnosis in these cancers could be rooted in at least two possibilities: 1) decreased access to care overall while incarcerated or 2) life-long systemic injustice of patients who are of low socioeconomic status and make up the majority of the incarcerated persons in the United States. It must also be recognized that poor access health care is not only found in our prison system. It is perpetuated throughout each stage of life and leads to unstable households, adverse childhood events, unemployment, mental illness, substance use and acquisition of habits that increase risk for cancer (smoking, alcohol use, sexually transmitted infections, injection drug use and its associated infections). Therefore, we must promote screening of cancers in our incarcerated population but also recognize that prevention of patients becoming incarcerated starts with adequate access to health care early in life as well.”

Given the above, I don’t see how you can possible avoid at least some discussion about the health care available to this population. Is screening even available? Is any education provided? Don’t blame this on patient literacy—providers of this population (who have NO choices) have some responsibility for health education. How much does screening cost?

A follow-up study might be to determine if the prisons in the state have any cancer screening guidelines and how these are accessed.

We thank Reviewer #2 for their input and we have revised the manuscript to state: (line 294) – “Current practices for screening in our state’s prison system are unknown, however it is notable that none of the patients diagnosed with colorectal cancer or lung cancer had documented appropriate screenings while incarcerated or were referred to our hospital following a positive result from screening.”

---

## [Decision Letter · Decision Letter 1]

14 Jul 2020

PONE-D-19-34040R1

Cancer stage at presentation for incarcerated patients at a single urban tertiary care center

PLOS ONE

Dear Dr. Sunthankar,

Thank you for submitting your manuscript to PLOS ONE. After careful consideration, we feel that it has merit but does not fully meet PLOS ONE’s publication criteria as it currently stands. Therefore, we invite you to submit a revised version of the manuscript that addresses the points raised during the review process.

We look forward to receiving your revised manuscript.

Kind regards,

Sungwoo Lim, DrPH

Academic Editor

PLOS ONE

Reviewers' comments:

Reviewer's Responses to Questions

**Comments to the Author**

1. If the authors have adequately addressed your comments raised in a previous round of review and you feel that this manuscript is now acceptable for publication, you may indicate that here to bypass the “Comments to the Author” section, enter your conflict of interest statement in the “Confidential to Editor” section, and submit your "Accept" recommendation.

Reviewer #2: All comments have been addressed

2. Is the manuscript technically sound, and do the data support the conclusions?

Reviewer #2: Yes

3. Has the statistical analysis been performed appropriately and rigorously? 

Reviewer #2: I Don't Know

4. Have the authors made all data underlying the findings in their manuscript fully available?

Reviewer #2: Yes

5. Is the manuscript presented in an intelligible fashion and written in standard English?

Reviewer #2: Yes

6. Review Comments to the Author

Reviewer #2: Please see comments in attached PFD. I still feel that you lack appreciation for the lack of power that incarcerated people have in terms of accessing health care and for the great power that prison health care providers/system have for addressing providers.

7. PLOS authors have the option to publish the peer review history of their article (what does this mean?). If published, this will include your full peer review and any attached files.

Reviewer #2: **Yes: **Patricia J Kelly

---

## [Author Response · Author response to Decision Letter 1]

17 Jul 2020

1. Abstract - You are not examining potential delays, you are examining differences in stage at diagnosis. If you were examining potential delays, you would also be looking individual and system level reasons for delay.

We thank reviewer #2 for their input and our manuscript now refers to “differences” instead of “delays.”

2. Page 3, line 45 - At least put that they are not in a position to advocate for themselves first. Really, blaming poor health literacy on the part of prisoners for not getting screened is ridiculous. In almost every situation that I have seen--jails, prisons, county, state, federal, prisoners CANNOT advocate for themselves. I feel as if you are looking at this from only the perspective of a community provider or clinic. I am really trying to be non-judgmental about this but feel as if the whole writing team needs to spend a day shadowing a provider inside of a facility.

We thank reviewer #2 for their input and agree that these patients are unable to advocate for themselves while incarcerated and this is of great importance. We aimed to summarize the sparse research on etiologies of decreased screening, and did not intend to characterize differences in cancer staging as primarily a failure on the part of the incarcerated patient. The study team has extensive first-hand experience working with incarcerated populations. We agree that they are often limited by what they have available and often there are constrictions on practitioners ability to truly practice standard-of-care medicine with appropriate treatment of conditions (i.e. chemotherapy, hepatitis C treatment), let alone screening for these conditions and other preventative care. We have changed our introduction to now state: “Incarcerated patients (IP) face both systemic barriers to care as well as vulnerability due to impediments to self-advocating due to their status as an inmate. Both these issues may delay or limit their receipt of treatments that – outside the prison system – are more often given, such as direct-acting anti-viral agents for hepatitis C infection and opioid maintenance therapy for substance use disorders.” 

We have also removed a reference to lower health literacy that was on lines 92-94.

We have updated the sentence (line 99-101) to say the following “Together these studies have highlighted that IP are less likely to receive the appropriate screenings compared to the general population.”

3. Page 4, line 84 - So many qualifiers. Decreased availability, likely, also contribute. Just say it. Lack of availability contributes to lack of screening.

We have streamlined our introduction to now state: “Multiple reasons for decreased screenings among patients who are currently or previously incarcerated have been suggested, including poorer connection with primary care resources and lack of availability while incarcerated and following incarceration due to systemic barriers patients who were previously incarcerated also face.”

4. Page 14, line 284 - Encourage? Direct routine screening

We agree and have changed “encourage” to “direct” as recommended.

---

## [Decision Letter · Decision Letter 2]

28 Jul 2020

Cancer stage at presentation for incarcerated patients at a single urban tertiary care center

PONE-D-19-34040R2

Dear Dr. Sunthankar,

We’re pleased to inform you that your manuscript has been judged scientifically suitable for publication and will be formally accepted for publication once it meets all outstanding technical requirements.

Kind regards,

Sungwoo Lim, DrPH

Academic Editor

PLOS ONE

Reviewers' comments:

Reviewer's Responses to Questions

**Comments to the Author**

1. If the authors have adequately addressed your comments raised in a previous round of review and you feel that this manuscript is now acceptable for publication, you may indicate that here to bypass the “Comments to the Author” section, enter your conflict of interest statement in the “Confidential to Editor” section, and submit your "Accept" recommendation.

Reviewer #2: All comments have been addressed

2. Is the manuscript technically sound, and do the data support the conclusions?

Reviewer #2: Yes

3. Has the statistical analysis been performed appropriately and rigorously? 

Reviewer #2: I Don't Know

4. Have the authors made all data underlying the findings in their manuscript fully available?

Reviewer #2: Yes

5. Is the manuscript presented in an intelligible fashion and written in standard English?

Reviewer #2: Yes

6. Review Comments to the Author

Reviewer #2: Thanks for your changes. I feel strongly that we cannot only serve as the health care providers for this population, but must also acknowledge the systemic factors that contribute to their poor health outcomes.

7. PLOS authors have the option to publish the peer review history of their article (what does this mean?). If published, this will include your full peer review and any attached files.

Reviewer #2: No